# Impact of web accessibility on cognitive engagement in individuals without disabilities: Evidence from a psychophysiological study

**Merve Ekin**[1]*, **Krzysztof Krejtz**[1], **Carlos Duarte**[2], **Letícia Seixas Pereira**[2], **Ann Marcus-Quinn**[3], **Izabela Krejtz**[1]

**1** Institute of Psychology, SWPS University, Warsaw, Poland, **2** LASIGE, Faculty of Sciences, University of Lisbon, Lisbon, Portugal, **3** Faculty of Arts, Humanities and Social Sciences, University of Limerick, Limerick, Ireland

* mekin@swps.edu.pl

**Data availability statement:** Datasets from this study are available in the Zenodo repository,

## Abstract

Web accessibility features on websites are designed for individuals with disabilities that include low vision and cognitive impairments, but such features can benefit everyone. This study investigates the impact of accessibility features of the web on ambient/focal visual attention and cognitive processing in individuals without disabilities. The study involved 20 participants reading news websites with different levels of low vision and cognitive-related accessibility features while their eye movements and heart rate variability were monitored. The findings show that cognitive engagement declined over time when no accessibility enhancements were present. The study also demonstrates that enhancing cognitive accessibility leads to increased user cognitive engagement, while low vision accessibility features make websites easier to read. These findings are corroborated by self-reports and psychophysiological measures, such as eye-tracking metrics and heart rate variability. The effects from these psychophysiological measures, together with participants' self-reports, support the benefits of enhancing web accessibility features for all users. The implications for future website design are also discussed.

## Introduction

Accessibility barriers on websites can significantly impact people with disabilities, limiting their access to information and services [1,2]. However, research and practical experience have shown that improving web accessibility often benefits all users, not just those with disabilities [3,4]. For example, captioning in videos not only aids those who are deaf or hard of hearing, but also supports language learners and those in noisy environments [5]. Screen readers, while essential for blind users, also benefit those who prefer to consume content audibly [6]. These instances underscore the broader value of accessible design, aligning with the principles of universal and inclusive design [7–9].

accessible via DOI: https://doi.org/10.5281/zenodo.15342685.

**Funding:** This work was financially supported by the European Union through the Marie Skłodowska-Curie Actions Doctoral Network, Eyes for Interaction, Communication and Understanding – Eyes4ICU project (Grant No. 101072410; authors M. E., K. K. and I. K.) [https://marie-sklodowska-curie-actions.ec.europa.eu/] and by the Portuguese Foundation for Science and Technology (FCT - Fundação para a Ciência e Tecnologia) through the LASIGE Research Unit (ref. UID/00408/2025; authors C. D. and L. S. P.) [https://lasige.pt/]. The funders had no role in the design, conduct, analysis, or interpretation of the study.

**Competing interests:** There are no financial or non-financial conflicts of interest.

Although it is widely accepted that accessibility to content levels can benefit all users, this assertion has not been rigorously validated by empirical research [3,10,11]. Although accessibility research often includes participants without disabilities, their involvement is usually limited to roles as stakeholders, proxies, or comparison groups [12] rather than exploring how accessible content affects their user experience. Most studies on web accessibility have concentrated on technical and usability aspects, leaving a significant gap in our understanding of how accessible content impacts user experience more broadly [13].

The present study aims to address this research gap by examining the cognitive engagement experienced by users without disabilities when interacting with websites of varying accessibility levels. The examination will focus on two attentional patterns: ambient and focal processing. Ambient attentional processing refers to the broad attentional state of being aware of the surroundings, whereas focal attentional processing involves a selective focus on a specific task or a limited area of the visual field [29,30]. The visual attention stream can be conceptualized as a constant interplay between ambient and focal attention in response to task requirements and individual cognitive resources [14,32,66]. We modified websites on two dimensions: low vision accessibility and cognitive accessibility. Cognitive engagement is an individual's active involvement in information processing [15,16]. To measure cognitive engagement, we employ a multimodal data approach providing continuous and objective insights [17] into how people explore websites enhanced with accessibility features. We triangulate self-reports and psychophysiological methods – eye-tracking and heart rate variability metrics – to understand how different accessibility features influence cognitive engagement during website exploration.

To advance the understanding of accessibility in web design, this study makes three key contributions:

- **Impact of accessibility features on cognitive engagement:** We provide empirical evidence that web accessibility features improve attention and cognitive processing, enhancing cognitive engagement with web content for users without disabilities.
- **Use of innovative methodologies:** We triangulate advanced psychophysiological methods, including eye tracking and heart rate variability, to offer a detailed analysis of cognitive engagement with accessible content.
- **Practical implications for web design:** The findings suggest that integrating accessibility features into web design can enhance overall user cognitive engagement, which can benefit everyday digital consumption and education.

## Related work

To situate this study within the broader context, we review key works in web accessibility and biosignals in cognitive engagement, highlighting relevant findings that inform our research.

**The universal benefits of web accessibility.** Web accessibility is a crucial component of contemporary digital environments, ensuring that users with varying abilities can effectively access and interact with online content. Research consistently demonstrates that web accessibility not only benefits users with disabilities but also enhances the experience for users without disabilities.

Schmutz et al. [3] explored the effects of implementing Web Content Accessibility Guidelines (WCAG) on users without disabilities. Their study revealed that higher levels of accessibility led to improved performance and user satisfaction, indicating that accessibility features can enhance the overall usability of websites. This finding underscores the universal benefits of web accessibility, suggesting that designing with accessibility in mind can create a better

user experience for everyone. Features like clear text, good contrast, and logical navigation structures are universally beneficial [18].

Similarly, Vollenwyder et al. [19] investigated the impact of using plain and easy-to-read language on websites. They found that such language not only aids users with cognitive disabilities but also improves comprehension and navigation for users without disabilities. This highlights the dual benefits of accessible design choices, which can simplify content and make it more user-friendly for a broader audience.

Yesilada et al. [20] identified common barriers experienced by mobile users without disabilities. For instance, mobile users often face challenges with small screen sizes and dynamic menus that can be hard to navigate, similar to the difficulties encountered by people with low vision or motor impairments when navigating websites on desktops or laptops. Additionally, Schmutz et al. [21] demonstrated that higher accessibility levels positively impacted users without disabilities across different age groups and devices. Their study showed that accessibility features, such as clear navigation and readable content, enhance the user experience universally, regardless of the user's abilities or the device used.

A systematic literature review by Campoverde-Molina et al. [10] of educational websites revealed that, while these sites often fail to meet accessibility standards, the implementation of such standards could improve the overall user experience. The review highlighted that web accessibility benefits not only people with disabilities but also elderly users whose abilities may have declined with age. This indicates that features like keyboard shortcuts and simplified navigation can make web interactions smoother for all users.

Research consistently shows that accessibility features not only support users with disabilities but also enhance the overall user experience. However, while the studies reviewed showed benefits for all users, they were few and were based on perceived attributes, not quantitative physiological ones.

**User cognitive engagement - eye movements.** Online content providers seek to attract users' attention by keeping their emotional, cognitive, and behavioral engagement, fostering a connection between a user and content [22]. Focused attention is considered one of the most important aspects of cognitive engagement [23,24], which sometimes may even lead to a loss of awareness of the outside world and distortions in the subjective perception of time [24].

Lagun and Lalmas [23] note that eye-tracking metrics provide reliable indicators of user engagement. Following the *eye-mind assumption* by Just and Carpenter [25], a large body of literature showed that gaze features such as eye fixation duration, total fixation time (see review [26]) or pupil dilation [27] are related to the depth of perceived information processing. Two types of eye movements, fixations and saccades, play a crucial role in the analysis of the attention process. Fixations are the small spatial disparity eye movements whose main function is to stabilize the perceived image on the retina for further cognitive processing and saccades are the large scale eye movements with the primary function of relocating ambient/focal visual attention to the different point of visual field. The longer the fixation duration and total fixation time the more cognitive processing of information is involved, indicating a higher cognitive engagement. A review showed that pupil dilation could be an index of effort in cognitive control tasks [27], suggesting that pupil size is also associated with remembering and recalling information, demonstrating its role in information processing.

For example, Arapakis et al. [28] demonstrated that user gaze patterns vary depending on the level of interest in different news articles. They found that people spent significantly more time fixating on the titles of interesting articles than those of uninteresting articles. They also reported positive correlations between eye-movement metrics (fixation duration, total fixation duration, and fixation number) as indicators of focal attention and positive affect evoked by reading stimuli articles.

Dynamics of focal attention can be examined with the duration of fixations combined with the amplitude of saccades that follow these fixations [29–31]. Specifically, short fixations followed by long saccades are typical of ambient processing, while longer fixations accompanied by shorter saccades indicate focal information processing [31,32] suggesting higher cognitive engagement. Krejtz et al. [31] proposed a syntactic measure of ambient/focal processing (coefficient $\mathcal{K}$), based on fixation duration and the following saccade amplitude. The abscissa serves to indicate time, so that $\mathcal{K}$ acts as a dynamic indicator of fluctuation between ambient/focal visual attention patterns [30,31]. For example, focal attention measured with $\mathcal{K}$ was related to a better understanding of online lectures [33].

The level of focal attention is also related to the time spent on task [34]. Hopstaken et al. [35] observed a decline in attentional engagement and cognitive performance with increasing task duration, which they attributed to mental fatigue. Yu et al. [36] reported a decrease in pupil diameter and task performance over time. It has been shown that cognitive engagement in the reading task has task-related fluctuations [37,38]. They also showed that longer fixation times and better cognitive performance were associated with the task-relevant reading text. Nisiforou et al., [39] investigated the relationship between different cognitive styles, field dependent and independent, and website complexity. Their results showed that the field-dependent group, influenced by external visual cues, spent more time completing the task and had longer fixation durations and shorter saccades in the complex website task. These results emphasize the evolving nature of cognitive engagement over time and its connection to the complexity of the task.

According to motivational control theory [40], energy spent is controlled by intrinsic motivation during goal-oriented tasks. This can contribute to an understanding of the processes of cognitive engagement and its dynamics in tasks that require cognitive effort. We believe that enhancing websites with accessibility features designed to support individuals with cognitive disabilities and low vision can modify the dynamics of cognitive engagement indicated by eye-movement metrics including, dynamics of ambient and focal attention.

**Eye tracking and websites.** Eye-tracking studies of websites have been popular for diagnosing the effectiveness of designs by analyzing user attention patterns [41,42]. There is also a consensus that eye tracking is one of the most effective methods for measuring user attention during interaction with websites [43–45] or interactive systems in a broader perspective [46–49]. Interestingly, Alt et al. [49] proposed an interaction technique where the news website's adaptation depends on the user's gaze behavior.

The eye-tracking method has been successfully applied in various contexts of website design. One notable application is in online shopping and personalization [50], where it helps understand user preferences and behaviors. Additionally, it has been used to assess attention to privacy information [45], ensuring users are aware of their data security. Research has also explored age differences in website browsing [51], revealing how different demographics interact with online content. Eye-tracking can also aid in risk assessment by analyzing where users focus their attention on privacy information presented on student loan websites [52]. Boarman et al. [50] provided insights into the attention and behavior of consumers during online fashion shopping, suggesting that personalization and customization features resulted in higher attention. Despite research on various topics, eye-tracking studies in website accessibility are relatively scarce.

**User cognitive engagement - heart rate variability.** Heart rate represents the number of heartbeats per minute. Heart rate variability (HRV), defined as the variation in the time interval (ms) between sequential heartbeats (Inter Beat Interval – IBI), is an index of autonomic control of the heart [53]. Following standards in the HRV literature [54], we analyzed HRV by

calculating two time-domain measures, the mean interval and the standard deviations of IBI values, known as the beat-to-beat interval or RR interval, for each participant [55,56].

HRV is seen as effective as eye-movement metrics for detecting information processing, as HRV correlates with cognitive functioning [53]. However, depending on the mode of HRV recording, passive (resting HRV measured before or after a task) [53] or active (during active task performance) [56,57], different results have been obtained.

Studies that compared passive (non-task related) HRV, as in the systematic review by Forte et al. [53], demonstrated that enhanced performance on cognitive tasks is positively correlated with elevated resting heart rate variability (measured before task). According to Forte et al. [53], several studies reported lower resting HRV being related to weaker cognitive functioning. For example, Williams et al. [58] observed that lower resting HRV predicted lower attention control. In line with this, Siennicka et al. [59] reported that the ability to maintain attention is associated with higher resting HRV.

When HRV was measured during task performance (active task-related HRV), Luque-Casado et al. [56] observed that HRV was sensitive to the requirements of sustained attention. They reported lower HRV values in the working memory task demanding most cognitive engagement compared to the attention required during the performance of less cognitively demanding tasks (a vigilance task or a discrimination task). They also demonstrated that HRV decreases with time-on-task, which may indicate more focal processing or mental fatigue. A systematic review of task-related HRV suggests that with increased time-on-task, HRV may also increase, indicating mental fatigue [57]. This suggests that increased cognitive fatigue may result in cognitive disengagement from the task at hand.

To the best of our knowledge, there are no studies measuring resting or active HRV in relation to website accessibility. HRV metrics, such as the variability of heart rate inter-beat intervals, provide continuous and objective indicators of cognitive engagement. By analyzing time-domain heart rate variability, we aim to determine whether enhancing websites with accessibility features keeps users cognitively engaged with time-on-task. The analyses of heart rate variability will complement our understanding of how accessibility features impact interaction with websites.

## Present study

Previous research suggests that accessibility features facilitate digital content perception in groups with low vision [60] as well as among those with cognitive impairments [61]. There are fewer studies examining the role of accessibility features on digital content perception in users without disabilities. However, previous studies relied predominantly on subjective techniques, including self-assessment questionnaires and interviews [62], as opposed to utilizing physiological measurements derived from objective and real-time data.

In our work, we address this gap by triangulating eye-tracking, heart rate variability, and self-reports to provide evidence that accessibility features are beneficial for users without disabilities. To this end, we investigate the dynamics of ambient/focal visual attention and heart rate variability in response to digital content differing in accessibility levels. In addition, we accounted for individual differences in cognitive resources by including working memory capacity as a covariant in our analyses. Working memory is known to play a critical role in sustaining attention and managing cognitive engagement during mental tasks, including reading and information processing [63,71].

For our interaction with digital content, we chose two leading news portals, The New York Times (NYT) and BBC News. The original websites were modified to incorporate accessibility enhancements for low vision, such as adjustments to letter, word, and line spacing, as well as

features for cognitive impairments, including the use of simplified language and the removal of distracting content. The enhanced websites were evaluated in comparison to their original, unmodified versions.

Although previous research has shown that accessibility features can improve usability for diverse populations, most studies have focused on subjective user experience. In this study, we employed physiological measurements to evaluate cognitive engagement during interaction with websites enhanced with low vision and cognitive accessibility features. To the best of our knowledge, there is no published study measuring the effect of website accessibility features on adults without disabilities triangulating multimodal data of self-assessment, eye-tracking, and heart rate measures. By focusing on eye metrics, heart rate variability, and self-report questions, our hypotheses directly examine whether improvements in website accessibility support deeper attention and more sustained information processing.

One of the contributions of the present study is the empirical demonstration of how accessibility features help users focus their attention on digital content for a longer time and reduce their cognitive effort. Another contribution of the present study is the methodological approach employed to understand the impact of accessibility features on cognitive engagement, ensuring the reliability and objectivity of the method. Finally, integrating these accessibility features into digital environments could be useful for all users. With this understanding, we believe that digital content designers will be better equipped to incorporate accessibility features into website design, potentially promoting user engagement and enhancing overall user experience.

**Hypotheses.** The present study examines the impact of accessibility features on users' cognitive engagement during interaction with news website content. In general, we hypothesized that cognitive and low vision accessibility enhancements to the websites affect cognitive information processing differently, which led us to formulate two specific hypotheses.

**Hypothesis 1.** Cognitive accessibility enhancements help engage users' attention more deeply and for a longer time. The support for this hypothesis will come from the statistical analyses of eye movements, heart rate variability, and self-reported measures.

- **H1.1.** We hypothesize that the differences between the original and enhanced accessibility features of websites are reflected in comprehension accuracy and users' evaluations of content understandability.
- **H1.2.** We expect to observe differences in ambient/focal visual attention between the original websites and their enhanced versions. Referring to the classical *eye-mind assumption*, assuming a close relationship between fixation duration and depth of cognitive processing [25], we predict more focal attention indicated by the $\mathcal{K}$ coefficient as well as higher average fixation duration, total fixation time, and fixations count on websites with cognitive accessibility enhancements compared to websites without those enhancements, reflecting higher cognitive engagement.
- **H1.3.** We expect that cognitive accessibility would help maintain focal attention for a longer time during interaction with enhanced websites in comparison with the original sites.
- **H1.4.** We predict that accessibility features result in modified heart rate variability (HRV). The standard deviation of the beat-to-beat interval (RR interval) was the primary dependent variable for this hypothesis. Studies measuring task-related HRV [56] observed that with increased attention, HRV decreases.

**Hypothesis 2.** Low vision accessibility enhancements increase readability and the ease of information processing from news websites.

- **H2.1.** We expect differences in comprehension accuracy and readers' evaluations of website readability.
- **H2.2.** We expect differences in eye-movement metrics: shorter average fixations duration, shorter total fixation time, fewer fixations, and lower $\mathcal{K}$ coefficient in comparison to the unmodified versions of the websites.
- **H2.3.** If low vision accessibility features foster cognitive performance, we hypothesize longer average time intervals between sequential heartbeats (IBIs) when interacting with websites with low vision accessibility enhancements. Longer and more irregular time intervals between heartbeats may be indicative of ambient attention in information processing [55,64,65] which is more relaxing for cognitive resources [66].

## Materials and methods

This section outlines the approach used to evaluate the impact of web accessibility features on user engagement. Using psychophysiological tools such as eye-tracking and heart rate variability, we measured user interaction with different web content. The following subsections describe participants, websites' design, data collection, and the statistical methods employed to analyze cognitive engagement under varying accessibility conditions.

### Participants

Twenty non-native English-speaking participants were recruited to attend the study (from 15/10/2024 to 31/10/2024). All participants (12 females; average age $26.65 \pm 6.98$) were right-handed with normal or corrected-to-normal vision. The sample size was chosen based on methodological considerations, including the time-consuming nature of multimodal data collection and the within-subject design, which increases statistical power by controlling for individual variability.

### Preparation of websites for user testing

To evaluate the impact of the various accessibility features, two websites were selected and modified to create four distinct versions: (1) the original pages without enhancements, (2) pages with enhancements for low vision accessibility, (3) pages with enhancements for cognitive accessibility, and (4) pages with both low vision and cognitive accessibility enhancements.

The chosen websites were from the BBC and the New York Times websites (for parts of websites, see Fig 1). These websites were selected because they are globally recognized and widely accessed news platforms, ensuring familiarity and relevance to a diverse audience. Additionally, their design and content structures incorporate a variety of features representative of mainstream web accessibility and usability practices, making them ideal for examining the impact of accessibility modifications in a real-world context.

For the pages with low vision accessibility enhancements, the original pages were modified in order to ensure that they followed the guidelines of the Web Content Accessibility Guidelines (WCAG). Notably, we ensured that the websites (details in Table 1):

- Have color contrast conforming to the WCAG
- Do not rely on color alone to convey information
- Support adjusting the text size
- Have letter, word, and line spacing conforming to the WCAG
- Reflow the content when magnified

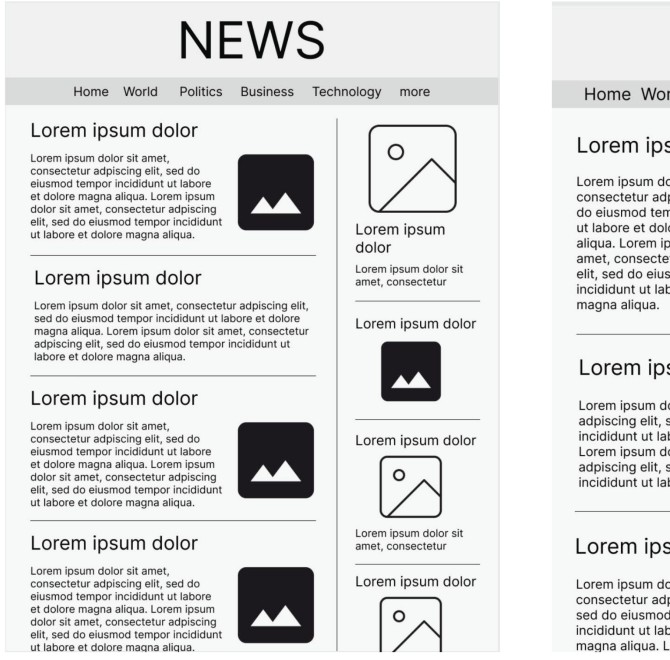

**(a)** Original

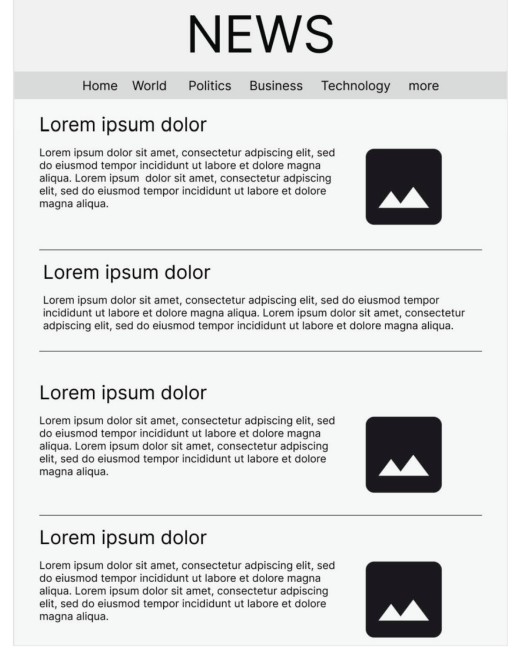

**(c)** Cognitive

**(b)** Low vision

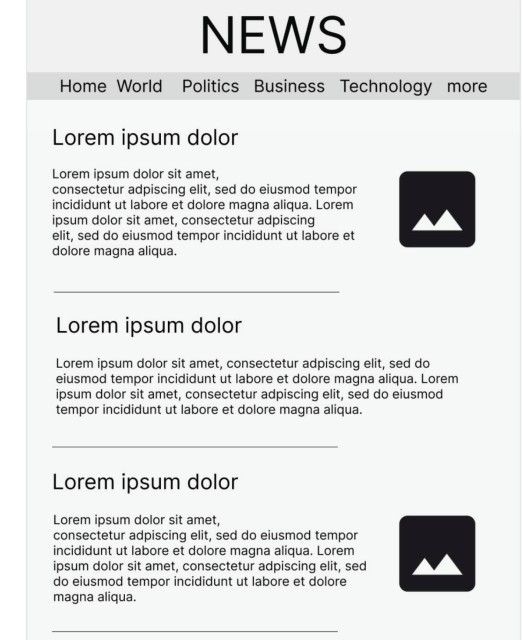

**(d)** Cognitive and low vision

**Fig 1. Mock-up version of the example of original New York Times website.** (1a) The original version of website. (1b) Low vision accessibility enhanced website. (1c) Cognitive accessibility enhanced website. (1d) Cognitive and low vision accessibility enhanced website. ***Note:*** *These figures were created by the authors for illustrative purposes. They are not identical to the original and have been recreated to align with the CC BY 4.0 license.*

For the pages with cognitive accessibility enhancements, the original pages were also modified with the same goal of following WCAG. In particular, we ensured that the websites (details in Table 2):

**Table 1. Low vision accessibility enhancements.**

| Enhancements | NYT page modifications | BBC page modifications |
|---|---|---|
| **Colour Contrast** | No changes needed as the page already conformed to WCAG standards. | No changes needed as the page already conformed to WCAG standards. |
| **Non-reliance on Colour Alone** | No changes needed. | No changes needed. |
| **Text Size Adjustment** | No changes needed. | No changes needed. |
| **Letter, Word, and Line Spacing** | Applied a line height of 1.5 times the font size, letter spacing of 0.09 times the font size, word spacing of 0.13 times the font size, and spacing after headings of 2 times the font size. | Applied a line height of 1.5 times the font size, letter spacing of 0.04 times the font size, word spacing of 0.16 times the font size, and paragraph spacing of 1 times the font size. |
| **Content Reflow** | No changes needed. | No changes needed. |

**Table 2. Cognitive accessibility enhancements.**

| Enhancements | NYT page modifications | BBC page modifications |
|---|---|---|
| **Simplified Language** | Summarized the text, reducing the readability grade from the original 16 to 10, and provided an option for users to read the original text. | Utilized the Hemingway App (https://hemingwayapp.com) to reduce the readability grade from 12 to 9, simplifying sentences while maintaining the integrity of quotes. |
| **Visual Aids** | No changes needed. | No changes needed. |
| **Avoiding Distracting Content** | Moved images within the news piece to after the content. | Removed sidebar images, moved ads and related links to after the content. |
| **Reading Time** | No changes needed. | No changes needed. |
| **Simplified Layouts** | No additional changes beyond those made to avoid distracting content. | No additional changes beyond those made to avoid distracting content. |

- Use simplified language
- Display visual aids to supplement written information
- Avoid distracting content
- Give enough time for users to read information
- Have simplified layouts

These modifications ensured that the websites were prepared to systematically test the impact of different accessibility enhancements, focusing on low vision and cognitive accessibility. Nevertheless, it is important to note that both pages were already quite accessible without the enhancements, particularly with regard to low vision accessibility.

## Self-report questions

Participants were initially introduced to the versions of the NYT page, followed by the versions of the BBC page. However, the four versions of each website were presented in a randomized order. Each version of the website was presented for 60 seconds. After interacting with each version of each website, participants were asked to answer three questions about (1) the article on the website, (2) the understandability of the website, and (3) the readability of the website (see questions in Table 3). The question about the content of the article was presented for 45 seconds, while the questions about readability and understandability were presented for 15 seconds each. Each participant navigated the websites and answered the questions within the same time frame to ensure consistency across all participants. The presentation time of the websites and questions was limited due to the potential impact on the reader's cognitive engagement if the experiment were extended, regardless of accessibility features.

**Table 3. Questions on website readability, understandability, and article comprehension.**

| Category | Question |
|---|---|
| Readability | How easy was it to read the content? |
| Understandability | How easy was it to understand the content? |
| NYT 1 | Why have Turkey and Hungary been hesitant to approve Sweden's NATO membership? |
| NYT 2 | What was the main topic of discussion during the meetings of Western nations, including NATO foreign ministers, in Norway and Moldova? |
| NYT 3 | What are the two main areas of focus for Western countries in relation to Ukraine's security and potential NATO membership? |
| NYT 4 | How has Russia's invasion of Ukraine affected the relationship between Western nations and NATO? |
| BBC 1 | In what way does the Galleri test identify cancer in individuals? |
| BBC 2 | What percentage of the 5,000 participants in the major NHS trial were found to have cancer through the multi-cancer blood test? |
| BBC 3 | What is the primary benefit of the Galleri test, as mentioned by the lead researcher, Prof Mark Middleton? What is the potential plan for the Galleri test in the NHS in England if initial results are successful? |
| BBC 4 | What percentage of the 5,000 participants in the major NHS trial were found to have cancer through the multi-cancer blood test? |

The pre-established questions related to the website article were also presented to the participants in a randomized order. There were four multiple-choice options, and participants provided the answer verbally to the researcher. For the understandability and readability questions, the participants answered on a scale from 1 (easy) to 7 (hard), which they also verbalized.

## Apparatus

The Gazepoint GP3-HD eye tracker with a sampling rate of 150 *Hz* was used to measure eye movements. Heart rate variability was recorded by the Gazepoint Biometrics - Finger Sensor Module. The devices were synchronized so that all physiological data could be recorded at the same time. The display screen used for the task presentation was 21.5 inches, with a 60 Hz refresh rate, and 1920×1080 screen resolution.

## The visual digit span task (visual DSPAN)

Working memory capacity was measured by visual digit span task [68,69]. In this task, participants saw forward and backward digit sequences (starting from 3 digits) for one second and recalled them by choosing the digits from a circle of digits from zero to nine with the mouse. If they gave a correct answer, the length of the sequence was increased, otherwise, the length of the sequence remained the same. After completing the 14 trials for the forward and backward conditions separately, the task was over. The task took approximately 15 minutes to complete and was administered using Inquisit 5 lab [70]. If participants recall all digits in the correct order, a sequence is scored correctly. In the analysis, the backward DSPAN score was chosen as an indicator of working memory because it typically requires more working memory capacity, suggesting a possible measure of cognitive engagement [71].

## Procedure

Participants gave written informed consent and were briefed on the study before performing the visual DSPAN task to assess their working memory capacity. All information was collected anonymously. Each participant then sat at a distance of 570 mm from the eye tracker,

placing their left middle and ring fingers on a finger sensor. A 5-point calibration and validation process was conducted to map gaze positions before starting the task.

The participants were presented with the websites for 60 seconds. Subsequently, the comprehension question was displayed for 45 seconds, while the understandability and readability questions were displayed for 15 seconds in order to obtain the participants' responses. The duration of the presentation of web pages and questions was chosen based on the sensitivity of psychophysiological measures to time-on-task. The main reason was to avoid fatigue effects and time overload when considering the whole experiment. Previous studies using eye-tracking in web contexts have also used similar durations to assess attention and user engagement [72,73]. Another reason for the duration was to ensure experimental control and maintain consistency across participants by keeping the duration limited. The entire physiological measurement phase, including the presentation of websites and questions, took almost 20 minutes (see Fig 2).

The finger sensor was cleaned with an alcohol wipe before use by each participant. Ambient light levels were measured, with an average light level of 11 lux recorded for the participants. The study was approved by the Ethical Review Board at the Faculty of Psychology, SWPS University (decision no: 47/2023).

## Data processing

For the preprocessing of raw eye-tracking data into fixations and saccades, we employed a non-parametric speed-based algorithm. The algorithm estimates velocity thresholds per

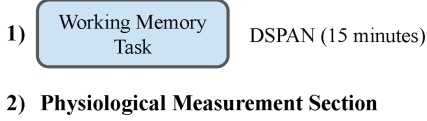

**1)** Working Memory Task  DSPAN (15 minutes)

**2) Physiological Measurement Section**

Web Page Task

Question 1

Question 2

Question 3

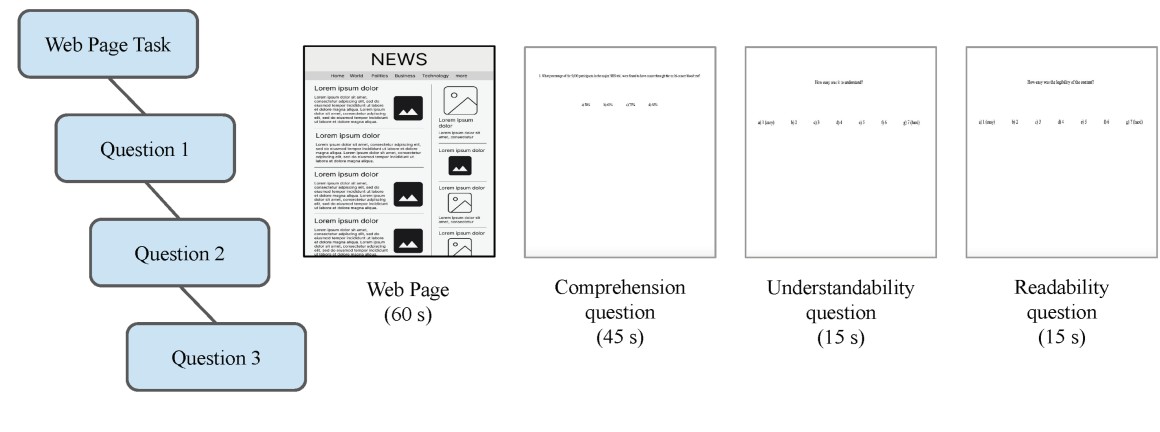

Web Page (60 s)

Comprehension question (45 s)

Understandability question (15 s)

Readability question (15 s)

**3)** Subjective Workload Task  NASA-TLX (~1-2 minute)

**Fig 2. Process diagram.** (1) The experiment began with the DSPAN task. (2) The physiological measurement phase was repeated eight times for all conditions (four NYT and four BBC) for each participant. First, the website was displayed for 60 seconds. Participants viewed the website in a browser and they were free to scroll and read the articles. Next, three questions appeared in a row to assess comprehension, understanding, and readability. Participants had 45, 15, and 15 seconds respectively to respond verbally. (3) The experiment concluded with the NASA-TLX task.

person and uses the duration lower threshold of 80*ms* for fixation duration. Additional consecutive fixations that overlap in space were combined [74]. Extreme fixation durations and saccadic amplitudes (over $M \pm 1IQR$) were trimmed to the highest non-outlying value.

The standard deviation of the RR intervals (beat-to-beat intervals), a standard time domain HRV measure [54], was calculated for HRV. Heart rate variability is recorded during the active task performance [57] in the present study. Inter-beat interval (IBI) values were calculated using R script which was based on a Matlab script provided by GazePoint Inc. The raw stream of the Heart Rate Pulse (HRP) is unitless but proportional to an ECG signal [75]. Local peaks were found in the HRP signal using the "findpeaks" function from the R signal processing package "gsignal" [76]. The distance between the peaks formed the inter-peak interval values, "known as the beat-to-beat interval or RR interval" [75]. Heart rate variability values were calculated as the standard deviation of the IBI values for each trial and each participant in the study.

The preprocessing was carried out using R computational language for statistical computing [77]. During the preprocessing stage, the following calculated: 1) heart rate variability metrics: the average inter-beat intervals (IBI) and standard deviation of IBI, beat-to-beat interval (RR interval); 2) eye-tracking metrics: fixation duration, total fixation time, fixation count, saccade amplitude, and $\mathcal{K}$ coefficient. The $\mathcal{K}$ coefficient introduced by Krejtz et al. [31] was obtained using the standardized *z*-score of fixation duration and saccade amplitude, and ambient/focal visual scanning was calculated per individual.

## Results

In this section, we present the results of our quantitative analysis, focusing first on the data, including self-reported metrics such as readability, understandability, and task performance. We then explore the findings from the psychophysiological measurements to understand how accessibility enhancements may influence cognitive engagement.

### Statistical analysis plan

The statistical analyses were conducted with the R language for statistical computing [77]. In order to test the research hypotheses, the analyses were divided into three series reflecting different metrics as dependent variables. The first set of analyses encompassed the accuracy of website content comprehension and self-reported questionnaire responses about the evaluation of website readability and comprehension. We conducted a series of $2 \times 2$ within-subject Analyses of Covariance (ANCOVAs) with low vision accessibility features (original website vs. enhanced website) and cognitive accessibility features (original website vs. enhanced website) as two independent within-subjects variables, and working memory capacity included as a covariate. To account for potential individual differences in working memory, the Backward Digit Span score was treated as a covariate in all our analyses.

The second set of analyses focused on scrutinizing the eye-movement metrics associated with the focus of attention, encompassing the second-order metric of $\mathcal{K}$ coefficient and the first-order metrics: number of fixations, average fixation duration and total fixation time as dependent variables.

The third set of analyses delved into the biomarkers of cognitive engagement, testing differences in the length of heart inter-beat intervals and heart rate variability as dependent variables. For testing the time-related hypotheses, in the second and third sets of ANCOVAs, we compared changes in eye-movement and HRV characteristics at the beginning of the website presentation with the end of the website reading time. To do that, in the ANCOVA model, we included a time epoch as the third independent variable leading to $2 \times 2 \times 2$ within-subjects

analysis design. The time epoch consisted of two levels (first vs. last 12 seconds of website viewing). The ANCOVA results are followed by pairwise comparisons with the HSD Tukey correction for multiple comparisons if needed, to decompose significant interaction effects. A *post-hoc* power analysis for each statistically significant effect was calculated to assess the sensitivity of the reported results. It was performed with the use of two R libraries *WebPower* and *effectsize*. Cohen's *F* and estimated power are reported alongside each result for the tested sample size *n*=20.

## Accuracy and self-assessment

Examining hypothesis 1.1, ANCOVA results on the self-assessment of understandability showed that websites with cognitive accessibility enhancements tended to be more understandable ($M = 4.89$, $SE = 0.197$) than websites without such enhancements ($M = 4.66$, $SE = 0.189$), see Fig 3a. However, this effect of cognitive accessibility enhancements reached only a statistical tendency level ($F(1, 18) = 3.40$, $p = 0.082$, $\eta^2 = 0.013$, Cohen's $F = 0.43$, $pwr = 0.45$).

Analogous ANCOVA for readability self-evaluation revealed a statistically significant main effect of low vision accessibility ($F(1, 18) = 8.96$, $p = 0.008$, $\eta^2 = 0.056$, Cohen's $F = 0.70$, $pwr = 0.84$), see Fig 3b. In line with predictions of hypothesis 2.1, websites with low vision accessibility features were evaluated as easier to read ($M = 5.02$, $SE = 0.156$) than websites without these features ($M = 4.60$, $SE = 0.180$). All other main and interaction effects were statistically insignificant, except for the covariant effect of working memory capacity ($F(1, 18) = 4.56$, $p = 0.047$, $\eta^2 = 0.127$). The following trend analysis showed that the higher working memory of the user the better the evaluation of readability in general ($\beta = 0.288$, $SE = 0.135$, $t(18) = 2.135$, $p = 0.047$).

The ANCOVA for accuracy of content comprehension showed no significant effects.

## Dynamics of visual attention

The analysis of ambient/focal attention dynamics depicted with coefficient $\mathcal{K}$ revealed three statistically significant effects (portrayed in Fig 4). First, the statistically significant main effect of cognitive accessibility features ($F(1, 18) = 7.53$, $p = 0.013$, $\eta^2 = 0.017$, Cohen's $F = 0.65$, $pwr = 0.79$). In line with the prediction of hypothesis 1.2, websites with cognitive

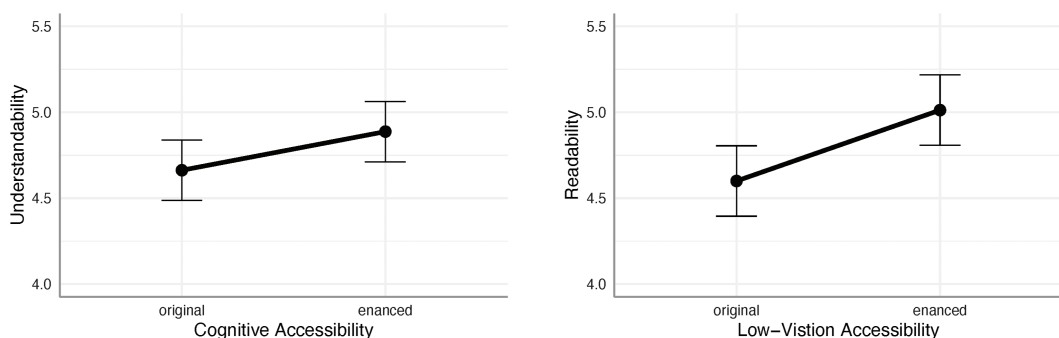

**Fig 3. Accessibility enhancements and self-report questions.** (3a) The main effect of cognitive accessibility enhancements on website understandability. (3b) The main effect of low vision accessibility enhancements on website readability. *Note:* The estimated means are depicted with dots; the whiskers represent lower and upper 95% confidence interval boundaries for the means.

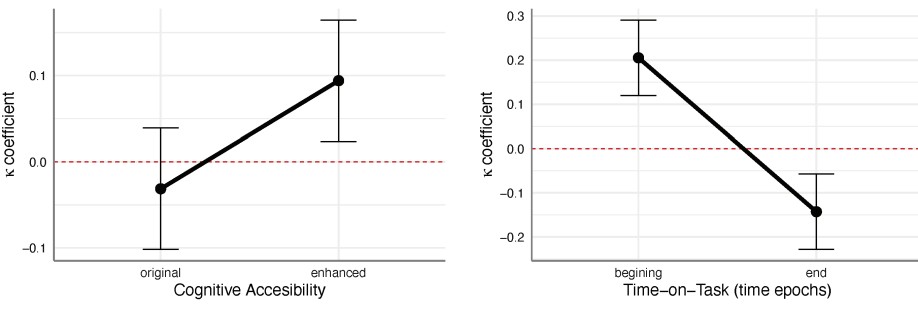

**(a)** Main effect of cognitive accessibility **(b)** Main effect of time-on-task

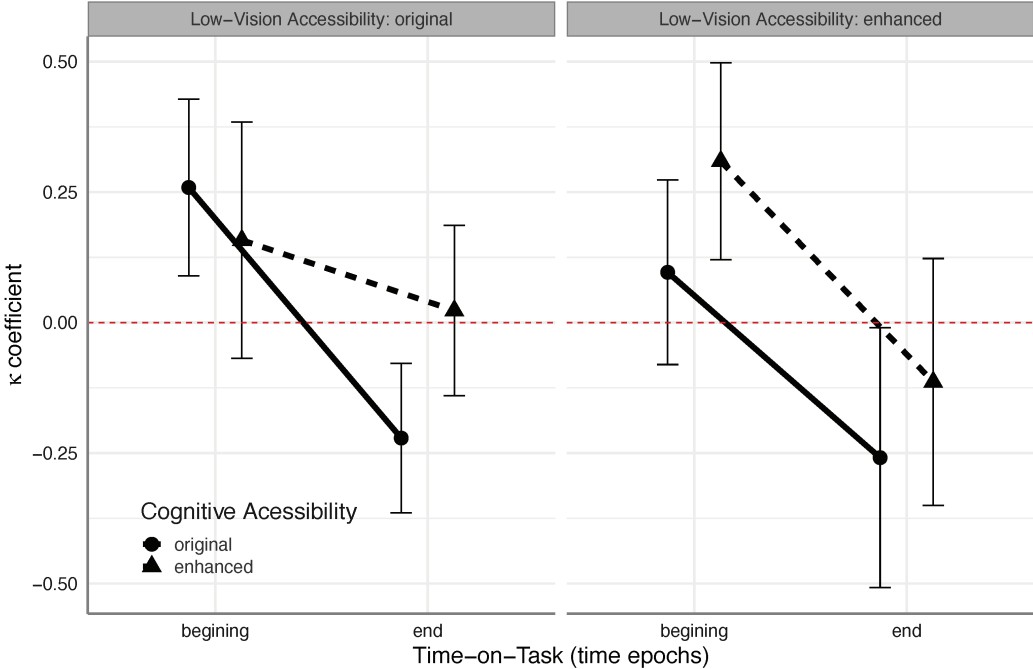

**(c)** Interaction effect of cognitive and low vision accessibility features.

**Fig 4. The influence of cognitive and low vision accessibility enhancements on focal attention dynamics during websites viewing.** (4a) Portraits the main effect of cognitive accessibility enhancements. (4b) Portraits the main effect of time-on-task. (4c) Portraits an interaction effect of cognitive and low vision accessibility enhancements. *Note:* The estimated means are depicted with dots and whiskers representing the lower and upper 95% confidence interval boundaries of the means.

accessibility enhancements, on average, yielded more focal attention ($M = 0.094, SE = 0.072$) than websites without these enhancements ($M = -0.031, SE = 0.072$), as displayed in Fig 4a.

Second, the main effect of time-on-task also reached the statistical significance level ($F_{(1, 18)} = 34.59$, $p < 0.001$, $\eta^2 = 0.117$, Cohen's $F = 1.39, pwr = 0.99$) showing that over time participants' attention was more ambient (hypothesis 1.3). At the beginning of the page viewing participants were more focal ($M = 0.205, SE = 0.082$), and more ambient attention at the end of website viewing ($M = -0.143, SE = 0.067$), see Fig 4b.

Third, the analysis showed a statistically significant effect of the three-way interaction of cognitive and low vision accessibility enhancements in time ($F_{(1, 18)} = 6.85$, $p = 0.017$, $\eta^2 = 0.012$, Cohen's $F = 0.62, pwr = 0.46$), see Fig 4c. The pairwise decomposition of this

**Table 4. Analysis of Covariance (ANCOVA) for the main effect of time-on-task on the first-order eye movement measures.**

| Eye-Movement Metric | Main Effect of Time-on-Task | Beginning Epoch | Ending Epoch |
|---|---|---|---|
| **Average Fixation Duration** | $F(1, 18) = 24.31, p < 0.001,$ $\eta^2 = 0.109,$ Cohen's $F = 1.16, pwr = 1$ | $M = 626ms, SE = 29.80$ | $M = 517ms, SE = 23.80$ |
| **Fixation Count** | $F(1, 18) = 40.80, p < 0.001,$ $\eta^2 = 0.001,$ Cohen's $F = 1.51, pwr = 1$ | $M = 32.90, SE = 2.95$ | $M = 32.70, SE = 2.95$ |
| **Total Fixation Time** | $F(1, 18) = 21.24, p < 0.001,$ $\eta^2 = 0.068,$ Cohen's $F = 1.09, pwr = 0.99$ | $M = 17822ms, SE = 1174$ | $M = 14822ms, SE = 902$ |

interaction effect showed that at the end, websites with cognitive accessibility enhancements and lacking low vision enhancements fostered more focal attention ($M = 0.023$, $SE = 0.099$) than websites without either cognitive or low vision accessibility enhancements in the same ending time epoch ($M = -0.221, SE = 0.089$), see Fig 4c, ($t(18) = -2.823$, $p = 0.011$). Interestingly, cognitive accessibility enhancements on websites without low vision accessibility adjustments protect users from attention loss (i.e., becoming ambient over time). The difference in coefficient $\mathcal{K}$ between first ($M = 0.158$, $SE = 0.139$), and last time epochs ($M = 0.023, SE = 0.099$), for those websites were not statistically significant ($t(18) = 0.93$, $p = 0.365$). However, corroborating hypothesis 2.2, there is a statistically significant ($t(18) = 2.15$, $p = 0.046$) drop of $\mathcal{K}$ towards an ambient attention between first ($M = 0.259, SE = 0.072$) and last time epochs ($M = -0.221$, $SE = 0.089$) on websites that were missing cognitive accessibility enhancements, see Fig 4c on left panel.

The results of the analyses of the first-order eye-movement-based metrics related to cognitive processing, average fixation duration, fixation count, and total fixation time revealed that the only significant effects were the main effects of time-on-task. The results of those three analyses are presented in Table 4.

## Heart rate variability and cognitive engagement

We obtained a statistically significant interaction between time-on-task and cognitive accessibility enhancements on websites, quantified by working memory capacity, ($F(1, 18) = 4.533$, $p = 0.047$, $\eta^2 = 0.025$, Cohen's $F = 0.50, pwr = 0.41$). The continuous nature of the working memory capacity covariate decomposition of this interaction effect requires the use of trends analysis (similar to simple slopes analysis) which allows for contrasting slopes of the relation between working memory capacity and HRV for each composition of independent variables levels (without cognitive accessibility enhancements in the first and last time epoch and with cognitive accessibility enhancements at the same time epochs). Test of hypothesis 1.4, such a *post hoc* analysis showed that working memory significantly predicts the HRV only for the last time epoch for websites with cognitive accessibility enhancements ($\beta = -0.512$, $SE = 0.212$, $t(18) = -2.415$, $p = 0.027$). That means that high working memory relates to higher cognitive engagement while reading websites in the last time epoch, but only when the website has been adjusted for cognitive accessibility. In other conditions, there was no effect of working memory on HRV. The trend analyses are presented in Fig 5.

When verifying hypothesis 2.3, the ANCOVA for heart rate variability as a dependent variable showed a marginally significant effect for low vision accessibility features ($F(1, 18) = 3.034$, $p = 0.097$, $\eta^2 = 0.026$, Cohen's $F = 0.41, pwr = 0.41$), showing marginally lower

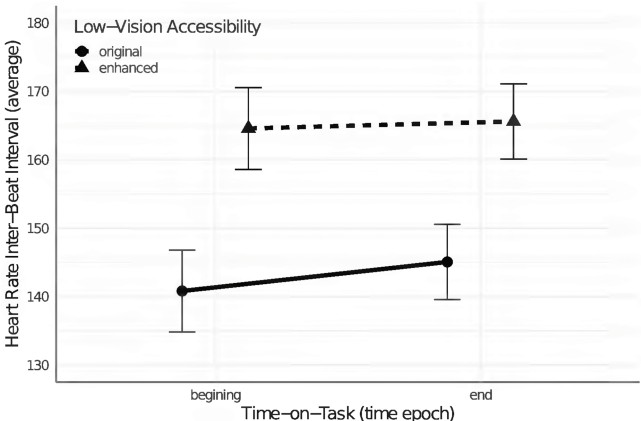

**Fig 5. Interaction effect of working memory capacity, heart rate variability and cognitive enhancement on the time-on-task.** The graph shows the relationship between users' working memory capacity and heart rate variability while reading websites with and without cognitive accessibility enhancements depending on the time-on-task epoch. Dot-dashed lines present slopes for the relationship between working memory capacity and HRV when reading websites with cognitive accessibility enhancement. Solid lines represent the slopes for the same relationship while reading websites without cognitive accessibility enhancements.

HRV for websites with low vision accessibility enhancements ($M = 0.155, SE = 0.124$) than for websites without those adjustments ($M = 0.481, SE = 0.124$). The analysis revealed the statistically significant effect of time-on-task ($F(1, 18) = 13.276$, $p = 0.002$, $\eta^2 = 0.050$, Cohen's $F = 13.28, pwr = 1$), indicating a significant increase in HRV between the beginning ($M = 0.088, SE = 0.007$) and the end of interaction with websites ($M = 0.548, SE = 0.126$).

Next, we verify the hypothesis 2.3. The ANCOVA with the mean heart rate inter-beat interval (average IBI) as a dependent variable revealed a statistically significant main effect of low vision accessibility enhancements ($F(1, 18) = 4.832$, $p = 0.0413$, $\eta^2 = 0.012$, Cohen's $F = 0.52, pwr = 0.60$) indicating that reading websites with low vision accessibility enhancements yields significantly longer IBIs ($M = 165, SE = 23.90$) than reading websites without these enhancements ($M = 143, SE = 21.40$). The analysis also showed a statistically significant main effect of time-on-task ($F(1, 18) = 256.942$, $p < 0.001$, $\eta^2 = 0.001$, Cohen's $F = 3.78, pwr = 1$) indicating an increase in IBI duration from $M = 153$ ($SE = 22.10$) at the beginning of website exploration to $M = 155$ ($SE = 22.20$) at the end of the website reading.

The above main effects were quantified by a significant interaction effect of time-on-task and low vision accessibility ($F(1, 18) = 9.845$, $p = 0.006$, $\eta^2 = 0.001$, Cohen's $F = 0.74, pwr = 0.77$). The pairwise comparisons suggest that the mean IBI for websites with low vision accessibility enhancements stays higher than for the original website. No significant change was observed for the mean IBI between the first and the last time epochs, ($t(18) = -1.998$, $p = 0.061$) (see Fig 6). For websites without low vision accessibility enhancements, there is a significant ($t(18) = -7.486$, $p < 0.001$) but a relatively small increase in duration IBI from $M = 141$ ($SE = 21.50$) in the first time epoch to $M = 145$ ($SE = 21.40$) in the last epoch.

## Discussion

The overall purpose of the present study was to understand the impact of accessibility features, specifically ones related to low vision and cognitive accessibility, on cognitive engagement for users without disabilities. In general, we hypothesized that cognitive and low vision

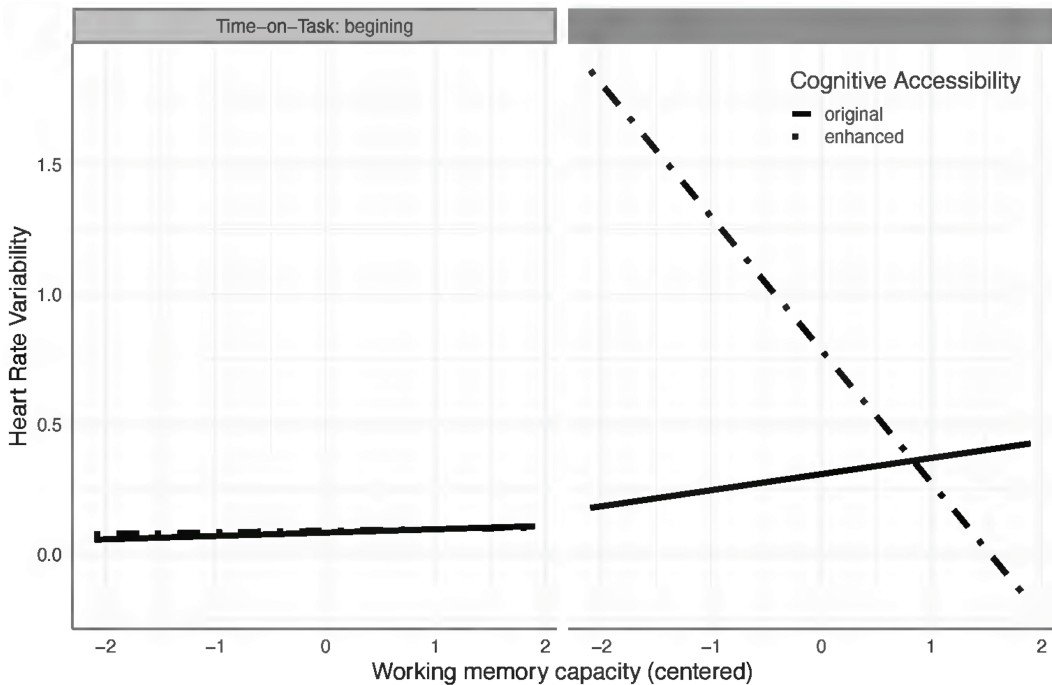

**Fig 6. The effect of time-on-task and low vision accessibility features on average heart rate inter-beat intervals during websites reading.** The graph shows effect of time-on-task and the average heart rate inter-beat intervals in low vision accessibility. Two lines represent different conditions of low vision accessibility. The solid line represents the original condition, showing a slight increase in heart rate variability from the beginning to the end of the task. The dashed line represents the enhanced condition. This line shows a stable and higher heart rate variability across the task with minimal change between the beginning and end.

accessibility enhancements to the websites will affect cognitive information processing differently. First, we expected that cognitive accessibility enhancements yield users' cognitive engagement (focal attention and thorough information processing). Furthermore, we expected that websites' accessibility enhancements promote keeping the cognitive engagement over time of website reading. Second, low vision accessibility enhancements were expected to facilitate the readability of the websites and the ease of information processing presented on them. Thanks to a unique methodological design, we intended to test those hypotheses using self-reports and psychophysiological measures based on eye movements and heart rate variability.

### The effects of cognitive accessibility on cognitive engagement (hypothesis 1)

In line with *hypothesis 1*, the present experiment results demonstrated that the depth of information processing from news websites depends on time-on-task and is moderated by cognitive accessibility enhancements made to the websites. In support of hypothesis 1.2, we observed that news websites enhanced with cognitive accessibility features were processed with significantly more focal attention than websites without these adjustments as indicated by $\mathcal{K}$ coefficient, suggesting higher cognitive engagement.

Our analyses also showed a decrease in cognitive processing depth over time during the reading of news websites. Values of eye-movement characteristics (average eye fixation duration, fixation count, and total fixation time, $\mathcal{K}$ coefficient) significantly decreased in time-on-task. The attention of the users, on average, was becoming more ambient (less focused), suggesting shallow cognitive processing of visual information. Similarly, previous studies [30,78] suggest time-dependent changes in cognitive engagement in the current task. Boksem et al. [79] found that task engagement (including reading and interaction with digital media) results in reduced attention over time, see also [78,80].

The decrease in cognitive engagement might be the effect of cognitive resource depletion over interaction time with the website. The present study revealed that website accessibility enhancements can stop this trend of cognitive engagement loss. We demonstrated that cognitive accessibility enhancements help users maintain focal attention until the last time epoch of website reading (hypothesis 1.3).

In partial support of hypothesis 1.4, the results revealed that cognitive accessibility enhancements are also beneficial for people with high cognitive resources. At the conclusion of the website reading tasks, we observed that lower heart rate variability, which indicates deeper cognitive processing, was associated with a higher working memory capacity among users. Shorter and more uniform time intervals between heartbeats (lower heart rate variability and shorter inter-beat intervals) may occur when someone is paying close attention. The present finding suggests that enhancing cognitive accessibility provides more consistent inter-beat intervals [81]. This can be interpreted as deeper information processing while reading websites enhanced with cognitive accessibility features [82,83]. Moreover, the websites enhanced with cognitive accessibility features were evaluated by users as more understandable, although this was a statistically marginal effect.

We may conclude that the incorporation of cognitive accessibility features, such as simplified language and the avoidance of distracting context, can enhance the user experience and facilitate the maintenance of user cognitive engagement (see also [84,85]).

## The effects of low vision accessibility enhancements on readability and the ease of cognitive processing (hypothesis 2)

As the cognitive accessibility enhancements yield focal attention when reading news websites, also low vision accessibility features facilitate the readability of the sites, making them easier to process, as predicted in *hypothesis 2*. In line with these predictions, self-report results showed that news websites with low vision accessibility enhancements are evaluated as easier to read (hypothesis 2.1). Although we did not find support for *hypotheses 2.2* in the eye movement-based data analyses, the heart rate related analysis corroborates results based on self-reports (hypothesis 2.3). We demonstrated that heart rate inter-beat intervals are significantly longer when reading websites with low vision enhancement, suggesting more relaxed (ambient) cognitive processing with lower effort. Longer and more irregular time intervals between heartbeats might be interpreted in terms of more ambient attention in information processing [55,64,65] which is more relaxing for cognitive resources [66]. We may conclude that the *hypothesis 2* related to low vision accessibility enhancement and their influence on the ease of information processing was partially supported.

Our study found no significant differences in reading comprehension between the original websites and the modified versions controlled for low vision and cognitive accessibility. This outcome may be attributed to the randomized order in which the page versions were presented to participants. The randomization could have led some participants to encounter the more understandable versions first, which may have facilitated their comprehension of

the content. As a result, this initial exposure likely enhanced their ability to answer questions about the article, regardless of the version they encountered afterward, thereby leveling the overall comprehension scores across different versions. Another potential reason might be that the limited task time forced participants to engage with the main ideas of the websites, regardless of the comprehension of the articles. While our psychophysiological measures detected statistical differences in cognitive engagement between conditions, the reading comprehension requires a longer sustained interaction with the content. Furthermore, the participants in this study were university students with high cognitive resources, which may have minimized the observed differences in reading comprehension between the original and enhanced website conditions.

## Innovative multimodal approach

This study distinguishes itself through the use of advanced psychophysiological methods, specifically eye-tracking and HRV measurements, to examine cognitive engagement in users without disabilities interacting with web content enhanced for accessibility. Unlike traditional self-report methods, which can be influenced by bias and subjective interpretation, these objective measurements provide continuous, real-time data on users' physiological responses [67]. Eye-tracking metrics, derived from fixation duration and saccade amplitude, offer precise insights into users' focal and ambient attention patterns while navigating websites, allowing us to quantify the level of cognitive effort and engagement elicited by different accessibility features. Meanwhile, HRV data provide a nuanced understanding of users' autonomic nervous system responses, capturing variations in cognitive load and attentional states over time. Together, these methods enabled a multidimensional analysis of how accessibility enhancements, such as simplified language and improved visual contrast, affect user experience in ways that would not have been evident through subjective measures alone [86].

By employing these innovative methodologies, the study uncovers several unique insights into the benefits of accessibility features. For example, the integration of eye-tracking data revealed that cognitive accessibility features, such as the use of simplified language and the removal of distracting elements, can sustain users' focal attention over extended periods, preventing the typical decline in cognitive engagement observed in standard web content [87]. This finding was further validated by HRV measures, which showed that these enhancements are associated with a reduction in cognitive fatigue, as indicated by more stable heart rate patterns during tasks. These physiological indicators provide robust empirical evidence supporting the idea that accessibility features do not merely facilitate ease of use but actively enhance cognitive engagement for a broader audience, including users without disabilities. Thus, the use of eye tracking and HRV establishes a new standard for studying web accessibility, moving beyond basic usability to a more comprehensive understanding of cognitive processes and user experience.

## Practical implications

The findings of this study offer several practical implications for web designers and developers, emphasizing the broad benefits of integrating accessibility features beyond compliance with legal standards [88]. The results demonstrate that both cognitive and low vision accessibility enhancements can significantly improve user engagement, even among users without disabilities. For example, features such as simplified language, reduced visual clutter, and enhanced contrast not only make web content more accessible but also maintain users' cognitive focus and reduce mental fatigue. These insights suggest that incorporating accessibility features can enhance the overall user experience, leading to longer engagement times and

potentially higher retention rates, which are critical factors for websites focused on content delivery, e-commerce, and education.

Based on the study's findings, several guidelines can be proposed to optimize web design. First, designers should prioritize the use of simplified language and structured content that supports cognitive accessibility, as these elements have been shown to sustain user attention. Avoiding distracting elements, such as intrusive ads and complex visual layouts, can further aid in maintaining focus. Second, designers should ensure adequate contrast ratios, customizable text sizes, and appropriate spacing in line with guidance for low vision accessibility. These adjustments not only support users with visual impairments but also make the content easier to read and process for all users, thereby broadening the appeal and usability of the website. By adopting these practices, web designers can create more inclusive online environments that cater to a diverse audience, potentially enhancing both user satisfaction and site performance metrics.

This study will be of significant interest to public sector organizations, particularly local government and the education sector. A recent study focusing on council websites in Spain found that although councils assert that their websites comply with accessibility standards, there has been minimal progress in achieving full accessibility. Enhancing web accessibility has the potential to boost citizen engagement at the local level while also improving well-being, credibility, transparency, and trust in local public institutions [89]. The number of individuals with disabilities enrolling in mainstream schools and higher education is steadily increasing [90]. Within the education sector, school and higher education institutions' websites are essential for supporting learning and disseminating information. EU regulations, along with the laws of many EU member states, mandate that public sector organizations, including schools, comply with WCAG 2.1 accessibility standards to ensure equitable access for all users. Despite these requirements, many schools and higher education websites still fail to fully meet accessibility standards, limiting access for some users [91]. The findings of this study will be valuable for web developers and designers of both school and college websites, offering insights and recommendations that could help improve the accessibility of these sites.

## Limitations and future directions

Overall, the present experiment has aligned with our expectations. However, it is not free from some limitations.

First, our participants were non-native English speakers interacting with English websites. Given the large volume of English content available on the Web, this is most likely an extremely common scenario, making it a valuable subject for investigation. As part of our adaptations, we simplified the content of the pages, and this change may, or may not, have a more pronounced impact on non-native speakers. Therefore, future studies may want to replicate the results on intralingual material. Accessibility features applied to websites in a foreign language may improve the understandability and readability of the content for non-native readers. In addition, the study used a within-subjects design to compare cognitive engagement with and without accessibility features, controlling for individual differences. We acknowledge that language background may influence the effects of accessibility features. Barnitz [92] emphasizes that reading in a second language is an interactive process shaped by the language proficiency, and cultural schema. This suggests that non-native readers may interpret and engage with content differently, potentially affecting their experience with accessibility features. Although our within-subjects design allowed us to control for individual differences, the role of language proficiency requires further investigation.

Second, two popular news platforms, the BBC and NYT websites, were selected for testing. Both pages were already relatively accessible, particularly for users with low vision. Despite this, we still observed significant results, suggesting the potential for an even greater impact on websites with lower accessibility levels. Future studies could explore these enhancements in more controlled environments with varying levels of accessibility to better understand their effects across different accessibility baselines.

Third, it can be noted that the relatively small sample size may have an impact on the generalisability of the results. We mitigate this limitation by applying a within-subjects design. To provide further information on the sensitivity of the outcome, the reported results were also assessed with a post hoc power analysis. Moreover, as far as we know, no previous studies have specifically examined the research question of the present study. However, further validation and a bigger sample are necessary to explore the broader impact of these findings on different groups of users.

Fourth, the duration of the presented material, websites and questions, was chosen based on the sensitivity of psychophysiological measures to time-on-task and to provide consistency across participants. However, longer durations may reflect more realistic web browsing behavior. Future studies could consider free browsing content in web design with accessibility features to assess cognitive engagement.

Fifth, the study focused on the impact of accessibility enhancements on cognitive engagement in users without disabilities. Future studies may want to replicate our findings on different samples. Further investigation may be conducted on enhanced websites examined by individuals with low vision and cognitive impairments. Testing the impact of cognitive and low vision accessibility features is also important from the perspective of the aging of society. Older adults observe a decline in cognitive and sensory functioning [93] therefore, this group may also benefit from a more inclusive and accessible digital content design. Exploring accessibility features designed for other types of disabilities, such as auditory, motor, or speech impairments, may also prove valuable. This broader inclusion ensures that the impact of improvements on individuals with disabilities and without disabilities can be compared.

## Conclusions

This study provides compelling evidence that accessibility features, such as cognitive enhancements and low vision adjustments, significantly improve cognitive engagement for all users, not just those with disabilities. Our findings demonstrate that web design practices focused on accessibility can enhance ambient/focal visual attention, sustain cognitive focus, and reduce mental fatigue for everyone, leading to a more inclusive and engaging user experience. By empirically validating these benefits through both subjective and objective measures, the study underscores the universal value of accessible design.

Theoretical implications of this research extend the understanding of web accessibility beyond its traditional scope, advocating for a more inclusive design approach that benefits diverse user groups. Practically, our findings offer concrete guidelines for web designers, such as using simplified language, minimizing visual clutter, and ensuring proper text contrast and spacing. These insights are relevant not only for enhancing user experience but also for informing policy and regulatory frameworks that govern digital accessibility standards.

Methodologically, this study is among the first to employ a combination of eye-tracking and heart rate variability measures to explore cognitive engagement in the context of web accessibility. This multi-modal innovative approach provides a richer, more nuanced understanding of how users interact with accessible content, offering a valuable template for future research in this field. The use of psychophysiological data allowed us to capture continuous,

objective insights into user behavior, revealing patterns of engagement that might have been missed by traditional survey methods alone.

In conclusion, this study contributes to the growing body of evidence that web accessibility features benefit all users, advocating for a shift toward more inclusive web design practices. By demonstrating that accessibility features enhance cognitive engagement and user readability, our research emphasizes the importance of integrating these features into mainstream digital environments. Ultimately, the findings promote a more accessible, user-friendly web for everyone, aligning with the principles of universal design and inclusive technology.

## Author contributions

**Conceptualization:** Merve Ekin, Krzysztof Krejtz, Carlos Duarte, Izabela Krejtz.

**Data curation:** Merve Ekin.

**Formal analysis:** Merve Ekin, Krzysztof Krejtz, Izabela Krejtz.

**Funding acquisition:** Krzysztof Krejtz, Izabela Krejtz.

**Methodology:** Merve Ekin, Krzysztof Krejtz, Izabela Krejtz.

**Project administration:** Krzysztof Krejtz, Izabela Krejtz.

**Software:** Merve Ekin, Krzysztof Krejtz, Izabela Krejtz.

**Supervision:** Krzysztof Krejtz, Izabela Krejtz.

**Visualization:** Merve Ekin, Krzysztof Krejtz, Carlos Duarte, Izabela Krejtz.

**Writing – original draft:** Merve Ekin, Krzysztof Krejtz, Carlos Duarte, Letícia Seixas Pereira, Izabela Krejtz.

**Writing – review & editing:** Merve Ekin, Krzysztof Krejtz, Carlos Duarte, Letícia Seixas Pereira, Ann Marcus-Quinn, Izabela Krejtz.

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
