## [Decision Letter · Decision Letter 0]

28 Mar 2025

PONE-D-25-08059Impact of Web Accessibility on Cognitive Engagement in Individuals Without Disabilities: Evidence from a Psychophysiological StudyPLOS ONE

Dear Dr. Ekin,

Thank you for submitting your manuscript to PLOS ONE. After careful consideration, we feel that it has merit but does not fully meet PLOS ONE’s publication criteria as it currently stands. Therefore, we invite you to submit a revised version of the manuscript that addresses the points raised during the review process.

We look forward to receiving your revised manuscript.

Kind regards,

Fredrick Romanus Ishengoma

Academic Editor

PLOS ONE

Journal Requirements:

5. We note that Figure [xxxx] includes an image of a [patient / participant / in the study].

6. We are unable to open your Supporting Information file [paper.tex]. Please kindly revise as necessary and re-upload.

Reviewers' comments:

Reviewer's Responses to Questions

**Comments to the Author**

1. Is the manuscript technically sound, and do the data support the conclusions?

Reviewer #1: Yes

Reviewer #2: Yes

2. Has the statistical analysis been performed appropriately and rigorously? 

Reviewer #1: Yes

Reviewer #2: Yes

3. Have the authors made all data underlying the findings in their manuscript fully available?

Reviewer #1: Yes

Reviewer #2: Yes

4. Is the manuscript presented in an intelligible fashion and written in standard English?

Reviewer #1: Yes

Reviewer #2: Yes

5. Review Comments to the Author

Reviewer #1: Thank you for the opportunity to read and comment on the paper, Impact of Web Accessibility on Cognitive Engagement in Individuals Without Disabilities: Evidence from a Psychophysiological Study. This study examines how web accessibility features designed for individuals with low vision and cognitive impairments can benefit all users. Using eye-tracking and heart rate variability measures, researchers analyzed the effects of these features on visual attention and cognitive processing in 20 participants without disabilities as they read news websites. The results indicate that cognitive engagement typically declines over time but can be maintained with cognitive accessibility enhancements. Additionally, low vision accessibility features improve readability. Both self-reports and psychophysiological data confirm the advantages of incorporating these features, highlighting their broader implications for inclusive web design.

Overall, I found the paper to be very well written, organized and structured in a clear manner with a logical flow. I appreciate the level of detail the authors provide in their hypotheses, methodological rigor, and statistical analysis plan. Their descriptions demonstrate rigor and transparency in their methods and analyses. I particularly found the analyses and results from the dynamics of visual attention to be a strong contribution from the study, as well as the mixed-methods and multimodal approach.

I only have some minor comments for the authors.

1. I have concerns regarding statistical power. I would like to see some explanation for how the sample size was chosen (e.g. a priori), and ideally, a power analysis. I found this lack of explanation surprising considering the level of statistical rigor in the rest of the text (it is not until the limitations section that the sample size is discussed).

2. It is briefly addressed in the limitations section, but I have some questions and would like further explanation from the authors regarding the sample consisting of non-native English speakers being examined on English text, and their statement: “we have no reason to suspect that the effects of accessibility features would have a different impact on native speakers”. I can think of several reasons why one might expect some different effects:

a. Studies have shown the use of subtitles, for example, differentially effects measures of realism, transportation, identification with characters, and other factors depending on if the subtitles are in ones own native language (e.g. Kruger, J. L., Doherty, S., & Soto-Sanfiel, M. T. (2017)

b. There are claims that second language reading is an interactive process, involving the interrelationship of cultural schemata and discourse structure – this could potentially have an impact on several of the measures included in this paper (see Barnitz)

c. There are even some specific findings on interpreting news media in non-native English speakers (Ward 2018), showing small but possibly relevant differences

The role that accessibility features might play in the above examples remains an open question – but I would like the authors to address this possibility or least better explain why they “have no reason to suspect” different impacts based on native language

Reviewer #2: The Introduction

Consider adding a brief discussion of the expected relationship between working memory capacity and cognitive engagement in the Introduction section to strengthen the hypotheses.

Definitions of key terms like "ambient attention," "focal attention," and "cognitive engagement" should be introduced earlier in the Introduction section in order to improve reader understanding.

The hypotheses are mentioned towards the end of the section. While they are clear, they could be better integrated into the narrative flow. Explaining how these hypotheses directly address the identified research gap would strengthen the introduction.

Related Works

Related works provides a comprehensive literature review on web accessibility, cognitive engagement, and psychological measures. However, consider adding a section summarizing the research gap to improve coherence and align with the study’s objectives. The authors should discuss how their study builds upon or challenges the findings of previous research.

Materials and methods

The sample size (20 participants) is small, limiting generalizability. Provide a stronger justification for this sample size or address its limitations directly. The authors should discuss if the use of non-native English speakers interacting with English website could bring a variability in the results. Also, the description of participant characteristics could include more demographic details (e.g., educational background, cultural factors) to clarify sample diversity.

The time allocated for interacting with each website version (60 seconds) and answering questions (45 seconds for comprehension, 15 seconds for readability and understandability) seems relatively short and the authors should justify this duration and discuss any potential effects on the results.

Results and discussion

Results are presented logically with appropriate use of visual aids and statistical tests. While the results are presented clearly, the interpretation could be more detailed. The discussion of unexpected findings, such as the lack of significant differences in reading comprehension, could be expanded. The authors should speculate on potential reasons and implications more thoroughly.

Other shortcomings

The text lacks references to images.

The authors did not sufficiently address the assumptions of using the analysis of covariance.

6. PLOS authors have the option to publish the peer review history of their article (what does this mean?). If published, this will include your full peer review and any attached files.

Reviewer #1: **Yes: **Joshua Juvrud

Reviewer #2: No

---

## [Author Response · Author response to Decision Letter 1]

9 May 2025

Dear Reviewers and Editorial Board,

We would like to express our gratitude for your valuable comments and the opportunity to revise our manuscript according to your recommendations.

The manuscript has been thoroughly revised to address the reviewers' comments and enhance its technical and methodological quality.

We hope that the revised manuscript will be a suitable fit for the PLOS One Journal, and we would like to thank the reviewers for their support in achieving this.

Yours sincerely,

Merve Ekin

---

## [Decision Letter · Decision Letter 1]

3 Jul 2025

Impact of Web Accessibility on Cognitive Engagement in Individuals Without Disabilities: Evidence from a Psychophysiological Study

PONE-D-25-08059R1

Dear Dr. Ekin,

We’re pleased to inform you that your manuscript has been judged scientifically suitable for publication and will be formally accepted for publication once it meets all outstanding technical requirements.

Kind regards,

Agnese Sbrollini

Academic Editor

PLOS ONE

Additional Editor Comments (optional):

Reviewers' comments:

Reviewer's Responses to Questions

**Comments to the Author**

1. If the authors have adequately addressed your comments raised in a previous round of review and you feel that this manuscript is now acceptable for publication, you may indicate that here to bypass the “Comments to the Author” section, enter your conflict of interest statement in the “Confidential to Editor” section, and submit your "Accept" recommendation.

Reviewer #1: All comments have been addressed

Reviewer #2: All comments have been addressed

2. Is the manuscript technically sound, and do the data support the conclusions?

Reviewer #1: Yes

Reviewer #2: Yes

3. Has the statistical analysis been performed appropriately and rigorously? 

Reviewer #1: Yes

Reviewer #2: Yes

4. Have the authors made all data underlying the findings in their manuscript fully available?

Reviewer #1: Yes

Reviewer #2: Yes

5. Is the manuscript presented in an intelligible fashion and written in standard English?

Reviewer #1: Yes

Reviewer #2: Yes

6. Review Comments to the Author

Reviewer #1: =The authors have addressed all of my previous comments clearly and thoroughly. I have no further concerns, and I believe the manuscript is much improved. I appreciate the authors' careful attention to the feedback provided.

Reviewer #2: All of my issues have been addressed. I have no further comments. I recommend this paper for publication in this journal.

7. PLOS authors have the option to publish the peer review history of their article (what does this mean?). If published, this will include your full peer review and any attached files.

Reviewer #1: **Yes: **Joshua Juvrud

Reviewer #2: No

---

## [Editor Report · Acceptance letter]

PONE-D-25-08059R1

PLOS ONE

Dear Dr. Ekin,

I'm pleased to inform you that your manuscript has been deemed suitable for publication in PLOS ONE. Congratulations! Your manuscript is now being handed over to our production team.

Kind regards,

on behalf of

Dr. Agnese Sbrollini

Academic Editor

PLOS ONE